# HYPERBOLIC DEEP REINFORCEMENT LEARNING FOR CONTINUOUS CONTROL

**Omar Salemohamed** [*] [12]    **Edoardo Cetin** [4]    **Sai Rajeswar** [5]    **Arnab Kumar Mondal** [135]

[1] Mila    [2] Université de Montréal    [3] McGill University    [4] King's College London
[5] ServiceNow Research

## ABSTRACT

Integrating hyperbolic representations with Deep Reinforcement Learning (DRL) has recently been proposed as a promising approach for enhancing generalization and sample-efficiency in discrete control tasks. In this work, we extend hyperbolic RL to continuous control by introducing a novel hyperbolic actor-critic model. Empirically, our simple implementation outperforms its Euclidean counterpart, with significant gains on 16/24 tasks from the DeepMind Control Suite with pixel inputs. Notably, in the low-data regime, our method even outperforms several pre-trained unsupervised RL agents. Our findings show that hyperbolic representations provide a valuable inductive bias for continuous control.

## 1 INTRODUCTION

Deep Reinforcement Learning (DRL) has shown great promise in solving continuous control tasks in high-dimensional environments (Yarats et al., 2021; Lillicrap et al., 2015). However, one of the biggest challenges in this field remains learning meaningful representations of observational data that can be used to train a policy efficiently. Recently, several methods have been proposed to address this challenge, such as training DRL agents with auxiliary tasks (Jaderberg et al., 2016) and using data augmentation (Yarats et al., 2021). Despite these advancements, these approaches only encode representations in a Euclidean space, limiting their ability to capture the underlying structure of the data. In contrast, recent studies have demonstrated the potential of non-Euclidean spaces in providing a stronger inductive bias and improving sample efficiency in DRL (Cetin et al., 2022; Mondal et al., 2022). In this work, we incorporate hyperbolic geometry as an inductive bias to learn better representations for DRL, aiming to improve the agents' performance and sample efficiency in continuous control tasks.

The motivation behind using hyperbolic spaces for learning better representations in DRL stems from their ability to encode hierarchical structures, such as trees, with minimal distortion (Sarkar, 2012)[1]. Given that the evolution of a Markov Decision Process (MDP) can be structured as a tree, with states represented as vertices and the transition dynamics of the MDP defining edges between states, using a hyperbolic inductive bias may facilitate learning representations that hierarchically evolve alongside the MDP. By leveraging hyperbolic geometry in this way, we aim to improve the sample efficiency and overall performance of DRL agents trained on continuous control tasks. As naively adapting the method proposed by Cetin et al. (2022) for discrete control struggles to learn effective policies, we propose an alternative approach that outperforms its Euclidean counterpart across 24 tasks from the DeepMind Control Suite (DMC) (Tassa et al., 2018) and several pre-trained unsupervised RL agents (Laskin et al., 2021) in the low-data regime.

## 2 METHOD

We integrate our hyperbolic layer into an actor-critic agent — the dominant RL framework for continuous control (Schulman et al., 2017; Haarnoja et al., 2018). Specifically, we adapt DrQv2 (Yarats et al., 2021), an extension of DDPG (Lillicrap et al., 2015) that uses data augmentation for

---

[*] Correspondence to: omar.salemohamed@mila.quebec.

[1] As a full treatment of hyperbolic geometry and its applications to deep learning are well beyond the scope of this paper, we point interested readers to Ganea et al. (2018) and Nickel & Kiela (2017) for more information.

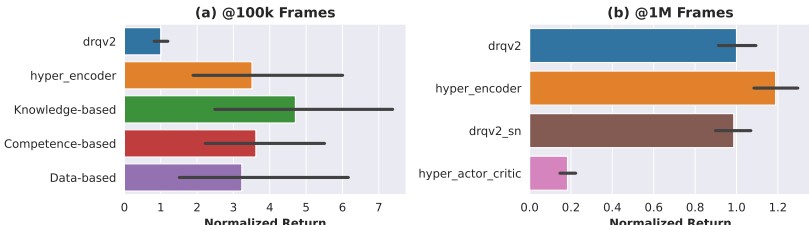

Figure 1: (a) Our **hyper_encoder** method compared with its Euclidean counterpart (**DrQv2**) and pretrained unsupervised RL agents evaluated on the URL Benchmark (A.3). (b) Average asymptotic normalized return across 24 DMC tasks. In addition to **DrQv2**, we compare our results with **hyper_actor_critic** (Cetin et al., 2022) and **DrQv2** with spectral normalization. On every experiment, we train each agent with 5 seeds. Results are normalized with the **DrQv2** performance.

better sample efficiency. Following Cetin et al. (2022), we first try two independent mappings to hyperbolic space[2], replacing the final layer of both the actor and critic networks. However, we find that this version (**hyper_actor_critic**) severely underperforms its Euclidean counterpart. Instead, we find placing the hyperbolic latent representation at the last shared layer of the feature encoder (**hyper_encoder**) to be the most effective approach. This change allows gradient signals from both the actor and critic models to optimize the hyperbolic representations, which we hypothesize might attenuate the noise magnitude in the initially noisy RL training signal. We provide results comparing the alternative architectures we explored in A.2. We also apply spectral normalization and rescaling to the convolution layers before the hyperbolic layer, following the *S-RYM* regularization procedure (Cetin et al., 2022). We find S-RYM helps mitigate training instabilities induced by the non-stationarity of the RL objective combined with the gradient instabilities inherent to employing hyperbolic representations (Ganea et al., 2018; López & Strube, 2020).

## 3 EXPERIMENTS

**Low-data regime (100k Frames).** We use the Unsupervised RL Benchmark (URLB) (Laskin et al., 2021) to compare our method in the low-data regime with unsupervised RL agents that have been pretrained for 500k frames. See A.3 for more details about the benchmark and the URL agents.

**Asymptotic experience regime (1M Frames).** In the asymptotic regime, we evaluate our method with 24 tasks from the DeepMind Control Suite (Tassa et al., 2018) with varying levels of difficulty. We provide an overview of the selected tasks in A.1 as well as task-specific results in A.3 & A.4. We compare our method to **hyper_actor_critic** Cetin et al. (2022) as well as two (Euclidean) baselines: DrQv2 and DrQv2 with spectral normalization. We include the latter as spectral normalization has also been shown to improve DRL agents(Gogianu et al., 2021). We use the default training and model hyper-parameters from Laskin et al. (2021), and for the hyperbolic layer, we use the hyper-parameters from Cetin et al. (2022).

As shown in Figure 1, the performance gains hold across the low-data and asymptotic regimes, validating that hyperbolic geometry provides DRL with a powerful inductive bias that facilitates recovering effective representations for continuous control. The results from the low-data regime are particularly surprising given that the URL agents have seen 5x more data during pre-training.

## 4 CONCLUSION AND FUTURE WORK

We introduce a novel actor-critic architecture that leverages hyperbolic representations and demonstrate its effectiveness for continuous control. Future work includes further investigating the learned representations' generalization capability and designing a hyperbolic intrinsic reward objective that incentivizes hierarchical exploration. These directions will help unlock the full potential of hyperbolic representations for DRL and advance our understanding of how to develop better DRL algorithms for complex environments.

---

[2]We implement the hyperbolic layer closely following Cetin et al. (2022), we point the interested reader to their work for a thorough description of its properties.

URM STATEMENT

The authors acknowledge that at least one key author of this work meets the URM criteria of ICLR 2023 Tiny Papers Track.

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

## A   APPENDIX

### A.1   DEEPMIND CONTROL SUITE TASKS

In Table 1 we provide an overview of the 24 dense and sparse-reward tasks from the DeepMind Control Suite (Tassa et al., 2018) that we use to evaluate our method. The Unsupervised RL Benchmark (URLB) (A.3) is composed of a subset of 12 of these tasks.

| Task | Traits | Difficulty | dim($Action\ Space$) | URLB |
|------|--------|-----------|----------------------|------|
| Cartpole Balance | balance, dense | easy | 1 | ✗ |
| Cartpole Balance Sparse | balance, sparse | easy | 1 | ✗ |
| Cup Catch | swing, catch, sparse | easy | 2 | ✗ |
| Finger Spin | rotate, dense | easy | 2 | ✗ |
| Pendulum Swingup | swing, sparse | easy | 1 | ✗ |
| Walker Stand | stand, dense | easy | 6 | ✓ |
| Walker Walk | walk, dense | easy | 6 | ✓ |
| Acrobot Swingup | diff. balance, dense | medium | 1 | ✗ |
| Finger Turn Hard | turn, sparse | medium | 2 | ✗ |
| Hopper Hop | move, dense | medium | 4 | ✗ |
| Quadruped Flip | flip, dense | medium | 12 | ✓ |
| Quadruped Run | run, dense | medium | 12 | ✓ |
| Quadruped Stand | stand, dense | medium | 12 | ✓ |
| Quadruped Walk | walk, dense | medium | 12 | ✓ |
| Reacher Easy | reach, dense | medium | 2 | ✗ |
| Walker Run | run, dense | medium | 6 | ✓ |
| Walker Flip | flip, dense | medium | 6 | ✓ |
| Humanoid Stand | stand, dense | hard | 21 | ✗ |
| Humanoid Walk | walk, dense | hard | 21 | ✗ |
| Humanoid Run | run, dense | hard | 21 | ✗ |
| Jaco Reach Top Left | reach, dense | hard | 9 | ✓ |
| Jaco Reach Top Right | reach, dense | hard | 9 | ✓ |
| Jaco Reach Bottom Left | reach, dense | hard | 9 | ✓ |
| Jaco Reach Bottom Right | reach, dense | hard | 9 | ✓ |

Table 1: A description of the 24 tasks used from the DeepMind Control Suite (Tassa et al., 2018). Table adapted from Yarats et al. (2021).

### A.2   HYPERBOLIC ACTOR-CRITIC ARCHITECTURE

We build our hyper actor-critic method by adapting DrQv2 which is built on top of DDPG and leverages data augmentation for better sample-efficiency. While we chose a DDPG backbone for ease of experimentation and sample-efficiency, there is nothing unique to our method with respect to this architecture and we believe our method can be easily integrated in a similar fashion to other on and off-policy actor-critic methods such as PPO (Schulman et al., 2017) and SAC (Haarnoja et al., 2018).

Following Cetin et al. (2022), we first try adding the hyperbolic layer to both the actor and critic networks. However, we find that this version (**hyper_actor_critic**) underperforms its Euclidean counterpart across all tasks, suggesting a different approach is necessary to design a hyperbolic agent for continuous control. Consequently, we experimented with using a hyper layer exclusively for the actor-head (**hyper_actor**), analogously for the critic-head (**hyper_critic**), and finally, placing it immediately after the CNN feature encoder (**hyper_encoder**), shared by both the actor and critic networks. While **hyper_actor** performed favorably to **hyper_actor_critic** and **hyper_critic**, we found **hyper_encoder** to be the best performing method. We compare the performance of these alternative architectures across 12 tasks from DMC in Figure 2.

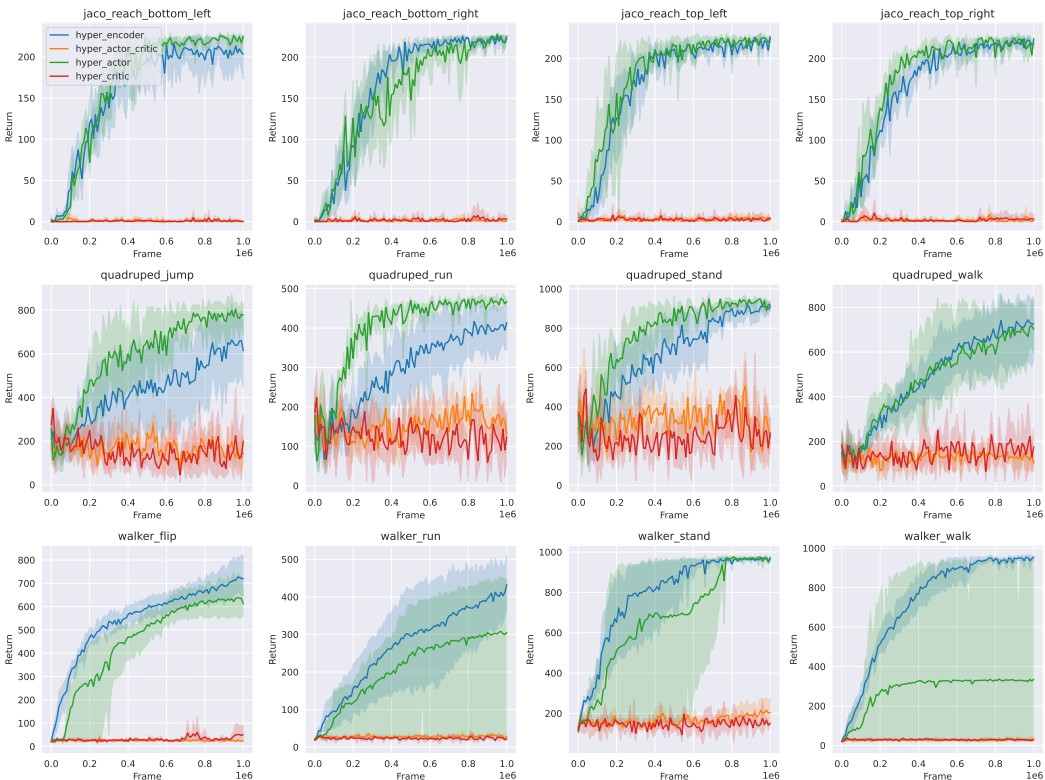

Figure 2: Performance curves of different hyper-actor critic architectures we experimented with.

## A.3 UNSUPERVISED REINFORCEMENT LEARNING BENCHMARK (URLB)

In order to better understand the extent of the performance improvement of **hyper_encoder** in the low-data regime, we compare our method to several unsupervised reinforcement learning (URL) baselines. We run experiments with the 8 agents presented in the Unsupervised RL Benchmark (Laskin et al., 2021) and refer to their work for an overview of each agent. Briefly, each agent has its own intrinsic objective that it leverages to perform unsupervised (reward-free) RL for up to 500k frames. For example, ICM (Pathak et al., 2017) learns a transition model of the environment which it utilizes to discover novel states in the absence of task-specific reward. After a reward-free pretraining stage, each agent is trained for 100k frames on a task *with* reward. This transfer setting has been shown to be an effective approach to improve sample-efficiency in the low data regime (Laskin et al., 2021).

Figure 3 plots the returns of each agent across the 12 tasks presented in URLB (see A.1 for an overview of the 12 DeepMind Control Suite tasks that compose URLB). In Figure 1(a), we use the same categorization scheme (Data-based, Competence-based, Knowledge-based) presented in URLB to group the different URL agents based on their instrinsic objective. However, we exclude the Disagreement, State Marginal Matching (smm), APT (icm_apt) agents from the statistics calculated in Figure 1(a) as our results show that these algorithms underperform the DrQv2 baseline trained from scratch.

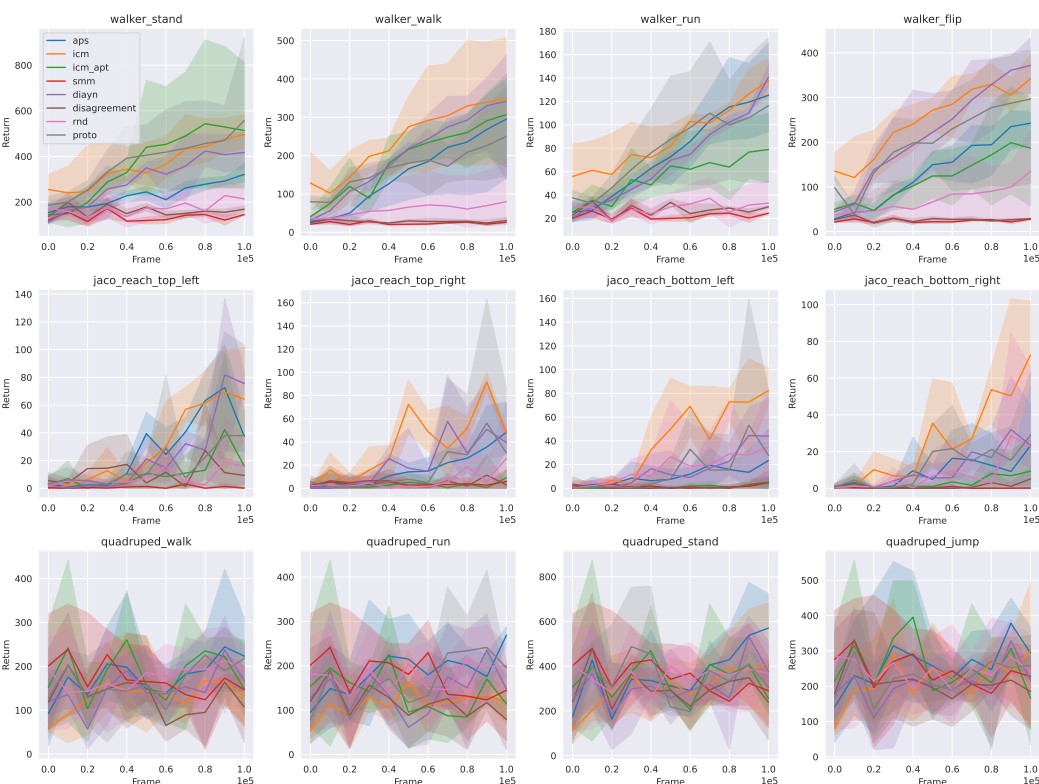

Figure 3: 100k performance curves of URL agents after 500k frames of task-agnostic training with intrinsic rewards.

## A.4 PERFORMANCE CURVES FOR 24 TASKS FROM DEEPMIND CONTROL SUITE TASSA ET AL. (2018)

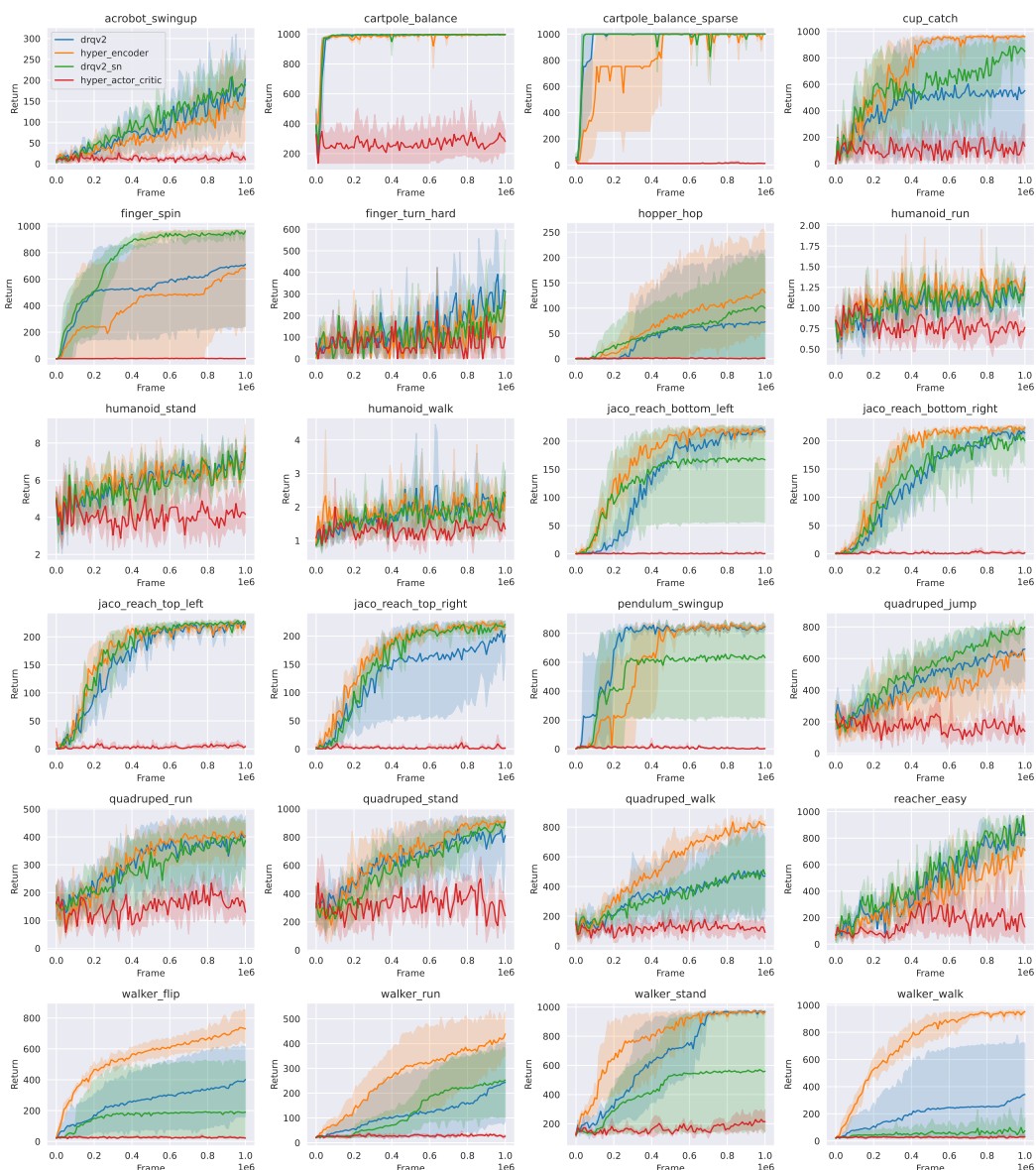

Figure 4: Performance curves of agents from Figure 1(b) after 1M frames of training.

