# OpenReview forum: "Hyperbolic Deep Reinforcement Learning for Continuous Control"
_ICLR.cc/2023/TinyPapers — Submitted to Tiny Papers @ ICLR 2023_

### Official Review · Reviewer_6eFF · 2023-03-26

**Confidence:** 4

**Summary Of Contributions:**

hard to identify the contribution due to writing issues

**Rating:**

Needs Clarification (NC): a submission which does not meet the reviewing criteria and needs clarification for its described problem or solution

**Strengths And Weaknesses:**

Summary:

The authors extend discrete hyperbolic reinforcement learning to continuous control tasks by integrating a hyperbolic layer into an actor-critic algorithm. The experiments have shown the proposed method outperforms its Euclidean counterpart, with significant gains on 16/24 tasks from the DeepMind Control Suite with pixel inputs.

Strength:

The experiments in the main text and appendix are comprehensive.

Weakness:

1. The paper style may not be suitable for this ICLR tiny paper track. Due to the page limit, the authors move the details of the hyperbolic layer to the appendix, making the paper not self-contained and may confuse the readers. Similarly, there is no explanation for the Euclidean counterpart, which is an important baseline in the experiments. Though page spaces are way too limited, the paper is supposed to be clear and self-contained.
2. There are some inaccurate understandings of basic RL knowledge: in the first sentence of the METHOD section, “an actor-critic agent — the dominant RL framework for continuous control” is not an accurate description. Actor-critic methods do be popular in continuous RL, but there is no such actor-critic agent thing. Because an actor actually represents an agent in AC methods.

**Suggested Changes:**

See my comments on weakness. The paper may not good for this tiny paper track. I would recommend adding more details and improving the writing, then submitting the revision paper to a regular paper track.

---

### Official Review · Reviewer_15qq · 2023-03-30

**Confidence:** 4

**Summary Of Contributions:**

The paper proposes an architecture that learns hyperbolic representations for continuous control. Through evaluations on the DeepMind Control Suite in a low data regime setting and asymptotic experience setting, they show that this inductive bias leads to an increase in performance across environments.

**Rating:**

High Potential (HP): a submission which meets the reviewing criteria and has potential to make an impact on the field

**Strengths And Weaknesses:**

## Strengths
- The paper well motivates the need for hyperbolic representations in deep RL. The paper is well written and clear.
- The proposed architecture has been compared against suitable baselines on a wide variety of environments.
    - In the experiments at 1M frames, HyperEncoder architecture improves performance over the base DrQv2 network architecture.
    - In the low data regime experiments, the hierarchical inductive bias introduced by HyperEncoder clearly helps DrQ improve its sample efficiency, nearly tripling the normalized return. The two settings strongly show the usefulness of hyperbolic representations.

## Weaknesses
- I am concerned by the size of error bars on the results in the low data regime. In some cases the error bars are nearly as large as the performance itself. More seeds should be run in these experiments to show that the performance increase is clear.
- Normalizing performance with respect to DrQv2 is a non standard way of comparing scores and I would prefer if the authors report scores normalized with respect to the max score in DMC (i.e. 1000).


**Suggested Changes:**

- Having additional seeds for the low data regime would be appreciated.
- I would have liked to see at least one other algorithm compared with and without the hyperbolic network architecture, for example PPO or SAC as mentioned by the authors. Given that the hyperbolic network architecture does not require any major assumptions on the algorithm, it should be easy to extend to an additional algorithm and it would further strengthen the validity of the authors’ claims.
- It would be better to aggregate performance across environments using the IQM score and Optimality Gap as proposed by Agarwal et. al 2021. This can be done easily using the [rliable](https://github.com/google-research/rliable) library released by those authors.

---

### Official Review · Reviewer_JN8G · 2023-04-02

**Confidence:** 4

**Summary Of Contributions:**

The paper uses hyperbolic spaces for learning better representation for DRL, extending prior work from discrete to continuous control tasks with a novel actor-critic methods.  The paper improves the sample efficiency and overall performance of DRL agents, trained both on URLB and DMC tasks.

**Rating:**

Clear, Correct, and Reproducible (CCR): a submission which meets the reviewing criteria

**Strengths And Weaknesses:**

### Strengths:
1. Great motivation for using hyperbolic spaces in continuous control tasks in RL.
2. The paper has thorough evaluations of both low-data and asymptotic experience regimes. It also clearly stated the model changes with respect to the simple architecture, which allows gradient signals from both the actor and critic models to optimize the hyperbolic representations.
attenuate the noise magnitude in the initially noisy RL training signal
3. The choice of the baselines is strong: hyper actor-critic Cetin et al. (2022) as well as two (Euclidean) baselines: DrQv2 and DrQv2 with spectral normalization. It's also interesting to see the comprehensive results on reward-free RL tasks.
4. Communication about contribution and generalization is clear. "While we chose a DDPG backbone for ease of experimentation and sample efficiency, there is nothing unique to our method concerning this architecture and we believe our method can be easily integrated in a similar fashion to other on and off-policy actor-critic methods such as PPO (Schulman et al., 2017) and SAC (Haarnoja et al., 2018)."

### Weaknesses:
Please refer to the next section for improvements.

**Suggested Changes:**

1. The paper compares the performance of alternative architectures across 12 tasks from DMC in Figure 2. It would be interesting to explain a bit about the performance differences across these tasks. Is it because of the dimensionality of the state/action spaces that impact the performance/variance?
2. The architecture and why it's beneficial is not entirely clear from the paper's limited description. It would be great to show more empirical details about the claim "hyper-encoder attenuates the noise magnitude in the initially noisy RL training signal" compared with traditional methods.

---

### Meta-Review · Area_Chair_Saow · 2023-04-07

**Recommendation:** Invite to archive
**Confidence:** 4

**Metareview:**

This paper is well motivated and well written. The contribution is clearly communicated. It would however benefit from more thorough, consistent experiments and thus might be more suitable to a venue where the page limit is not as tight.

**Summary:**

An innovative and effective actor-critic framework is designed to boost generalizability and sample efficiency of continuous control.

**Reason For Not Giving A Higher Recommendation:**

- multiple reviewers have pointed out that clarification is needed in some sections
- the paper could benefit from more thorough experiments

**Reason For Not Giving A Lower Recommendation:**

The paper is well-motivated and well-written.

---

### Decision · Program_Chairs · 2023-04-08

Invite to archive